# On the Reversible and Irreversible Inhibition of Rhodesain by Curcumin

**DOI:** 10.3390/molecules25010143

**Published:** 2019-12-30

**Authors:** Dietmar Steverding

**Affiliations:** Bob Champion Research and Education Building, Norwich Medical School, University of East Anglia, Norwich Research Park, James Watson Road, Norwich NR4 7UQ, UK; d.steverding@uea.ac.uk; Tel.: +44-1603-591291

**Keywords:** rhodesain, curcumin, non-competitive inhibition

## Abstract

Previously, it was suggested that the natural compound curcumin is an irreversible inhibitor of rhodesain, the major lysosomal cysteine protease of the protozoan parasite *Trypanosoma brucei*. The suggestion was based on a time-dependent inhibition of the enzyme by curcumin and a lack of recovery of activity of the enzyme after pre-incubation with curcumin. This study provides clear evidence that curcumin is a reversible, non-competitive inhibitor of rhodesain. In addition, the study also shows that the apparent irreversible inhibition of curcumin is only observed when no thiol-reducing reagent is included in the measuring buffer and insufficient solubilising agent is added to fully dissolve curcumin in the aqueous solution. Thus, the previous observation that curcumin acts as an irreversible inhibitor for rhodesain was based on a misinterpretation of experimental findings.

## 1. Introduction

Curcumin is a natural phenol and has been extensively investigated as potential drug candidate for various illnesses and medical conditions [1]. However, the compound has been classified as a pan-assay interference compound (PAINS) and an invalid metabolic panacea (IMPS) [1]. PAINS are compounds that show activity in different types of assay mainly through interfering with the assay itself while IMPS are reagents that display activity against virtually any biological target. Despite this and other drawbacks (chemical instability, low bioavailability, non-selectivity and toxicity), curcumin is still subject of intense research and about 50 papers are published each week on biological interactions of the compound [1].

Previously, it was shown that curcumin display anti-proliferative activity against the protozoan parasite *Trypanosoma brucei*, the causative agent of sleeping sickness in humans and nagana disease in livestock [2]. In search for the biological target involved in the trypanocidal activity of curcumin, the effect of the compound on rhodesain, the major lysosomal cathepsin L cysteine protease in *T. brucei*, has been recently investigated [3]. The enzyme is essential for the survival of the parasite and a valid drug target [4]. It was shown that curcumin was able to inhibit rhodesain and it was suggested that this inhibition was irreversible [3]. This conclusion was based on a weak non-linear relationship between substrate hydrolysis and incubation time (with increasing incubation time the hydrolysis of substrate decreased slightly) and the lack of recovery of enzyme activity after dilution of rhodesain pre-incubated with curcumin [3,5]. However, it was also recently shown that the inhibition of rhodesain by curcumin seemed to be reversible [6]. A 1:4 dilution of a reaction mixture containing rhodesain and 10 μM curcumin resulted in a 4.7-fold increase in activity [6]. In order to prove conclusively that curcumin is a reversible inhibitor, kinetic studies to determine the inhibitor type of the compound were carried out. In addition, further investigations were conducted to provide explanations for the apparent irreversible inactivation of rhodesain by curcumin recently observed [3,5]. The results of this study revealed that curcumin is a reversible non-competitive inhibitor of rhodesain, a new finding that disproves unequivocally previous claims that curcumin is an irreversible inhibitor. This study also showed that it is important to select the correct assay conditions to measure enzyme activity and to consider the solubility properties of inhibitors otherwise incorrect data will be obtained leading to a misinterpretation of results.

## 2. Results and Discussion

The activity of rhodesain was determined with the fluorogenic substrate benzyloxycarbonyl-phenylalanyl-arginyl-7-amido-4-methyl coumarin (Z-FR-AMC), a substrate that is cleaved by mammalian and trypanosome cathepsin L cysteine proteases [7,8].

Time course experiment revealed that the inhibition of rhodesain by curcumin was time independent. In the presence of 6 μM curcumin (a concentration close to the IC_50_ value for the inhibition of rhodesain by curcumin, see below), the inhibition of the activity of rhodesain was linear with respect to time (Figure 1). The correlation coefficient of the readings was 0.9997 confirming a strong linear association between substrate hydrolysis and incubation time. The same correlation coefficient was also determined for the control reaction (Figure 1) indicating that there was no difference in the linearity of the readings for the substrate hydrolysis in the presence and absence of curcumin. In contrast, when rhodesain was incubated with the established irreversible cysteine protease inhibitor CAA0255 [9] at 0.1 μM (a concentration below the IC_50_ value for the inhibition of rhodesain [4]), the activity of the enzyme was quickly completely inhibited (Figure 1 insert). Within 5 min of incubation, the activity of rhodesain was inhibited by >90%.

After establishing that curcumin is a reversible inhibitor of rhodesain (see above and [6]), kinetic studies to determine the inhibitor type of the compound were carried out. Double reciprocal analysis (Lineweaver–Burk plot) gave a family of lines with increasing slopes as the curcumin concentration increased (Figure 2a). The lines converged to the same point on the x-axis indicating a non-competitive inhibition mechanism (Figure 2a). Plotting the reciprocal velocity (1/v) against the inhibitor concentration (Dixon plot) gave again a family of lines that met in a single point on the x-axis confirming that curcumin is indeed a non-competitive inhibitor of rhodesain (Figure 2b). From the point of intersection, the apparent inhibitor constant K_i_ for curcumin was determined to be 5.5 ± 1.4 μM (n = 3).

As for a non-competitive inhibitor, the K_i_ value is equal to the IC_50_ value [10]; the IC_50_ value of curcumin for the inhibition of rhodesain was determined next. The compound inhibited the activity of rhodesain in a dose-dependent manner with an IC_50_ value of 5.6 ± 0.5 μM (n = 3) (Figure 3). The IC_50_ value was not statistically significantly different from the K_i_ value (Student’s *t* test; *p* = 0.971). This finding confirmed that curcumin is indeed a non-competitive inhibitor of rhodesain.

Having shown that curcumin is a reversible, non-competitive inhibitor of rhodesain, the question remains why previously a lack of recovery of activity after pre-incubation of the resting enzyme with the compound was found [3,5]. As the incubation of the active enzyme (i.e., in the presence of substrate) with curcumin is reversible [6], one could conclude that the compound binds with different affinities to the free enzyme and the enzyme-substrate complex. However, this explanation can be excluded as a non-competitive inhibitor binds equally well to the enzyme whether or not it has bound the substrate. A more likely reason for the observed apparent irreversible inhibitory activity of curcumin is the very low water solubility of the compound, which is just 0.6 μg/mL (1.63 μM) [11]. In this context, it is important to note that in the recent studies rhodesain was pre-incubated with 50–100 μM curcumin for 30 min before the recovery of the activity of the enzyme was determined by dilution of the reaction mixture into measuring buffer [3,5]. At concentrations of 50–100 μM, curcumin will be rather dispersed than dissolved in aqueous solutions. This notion is supported by previous observation that curcumin displays very low absorbance in aqueous solutions [12]. The dispersed curcumin particles may absorb and/or non-specifically inactivate rhodesain present in the reaction mixture. However, the water solubility of curcumin can be increased in the presence of DMSO (Figure A1). In order to check whether undissolved curcumin can non-specifically inactivate rhodesain, the enzyme was pre-incubated with 100 μM of the compound in the presence of DMSO at a low concentration of 0.1% and at a high concentration of 10%, respectively. After 30 min incubation, the reaction mixture was diluted 100-fold into measuring buffer containing substrate to give a curcumin concentration of 1 μM that was shown not to affect the activity of rhodesain (see Figure 3). The activity of rhodesain treated with curcumin in the presence of 0.1% DMSO was not fully restored after the dilution (Figure 4). It reached only 30% of the control enzyme activity. In contrast, the activity of rhodesain incubated with curcumin in the presence of 10% DMSO was restored to 91% of the control enzyme activity after the dilution (Figure 4). In this case, the activity of the treated enzyme was not statistically significantly different from that of the control enzyme (*p* = 0.337; Figure 4). This result shows that curcumin, if it is dissolved with the help of an appropriate solubilising agent, does not irreversibly inactivate rhodesain. Thus, the lack of recovery of curcumin pre-treated rhodesain observed recently [3,5] seemed to be most likely due to non-specific inactivation by undissolved curcumin particles present in the reaction mixture. Interestingly, a similar observation (lack of recovery of enzyme activity after pre-incubation with curcumin) was previously reported for the inactivation of CD13/aminopeptidase N [13]. While curcumin was identified as a reversible non-competitive inhibitor of CD13/aminopeptidase N, the activity of the enzyme pre-treated with curcumin could not be fully restored after three rounds of filtration using centrifugal filter devices to remove the compound.

Finally, the question remains as to why a time-dependent inhibition of rhodesain activity by curcumin was recently observed [3,5]. In this regard, it should be mentioned that the measuring buffer (assay buffer) used in the recent studies [3,5] did not contain any reducing thiol reagent. However, cathepsin L cysteine proteases are only fully catalytically active in the presence of thiol reagents (e.g., dithiothreitol, [14]). When determining the effect of curcumin on the activity of rhodesain in measuring buffer lacking dithiothreitol, a time-dependent inactivation of the enzyme activity by the compound was observed (Figure 5). After 30 min of incubation, the enzyme was almost completely inactivated. However, after addition of dithiothreitol to a final concentration of 2 mM, rhodesain regained its activity (Figure 5). Moreover, the activity of the enzyme was now linear with respect to time with a correlation coefficient of the readings of 0.9997 (Figure 5). This result clearly demonstrates that it is essential to include a thiol reagent in the measuring buffer in order to keep rhodesain fully activated. It should also be pointed out that the activity of rhodesain decelerated when measured in the absence of curcumin and dithiothreitol (Figure 5, insert). However, the time-dependent inactivation of rhodesain in the absence of dithiothreitol for the curcumin-treated enzyme was more pronounced than for the non-treated enzyme. These findings indicate that rhodesain in the absence of a thiol reagent is oxidised, which leads to gradual inactivation of the enzyme. This oxidation of the rhodesain seems to be accelerated in the presence of curcumin, which may be mistaken as an irreversible inhibition of the enzyme.

Other antioxidants (cysteine, glutathione, β-mercaptoethanol and ascorbic acid) were also able to reactivate rhodesain that had been inhibited by curcumin, although at different effectiveness (Figure 6). Cysteine was most effective in the reactivation of rhodesain (even better than dithiothreitol) while ascorbic acid could not sustain the reactivation of the enzyme in the longer term. Glutathione was a slow acting reagent but over time reactivated the enzyme to a similar extent as β-mercaptoethanol. In general, the effectiveness of the reactivation process of curcumin-inhibited rhodesain by the different antioxidants (cysteine > dithiothreitol > glutathione = β-mercaptoethanol > ascorbic acid) was determined by their standard redox potential E_0_’: the more negative E_0_’, the better the reactivation (E_0_’(cysteine) = −348 mV; E_0_’(dithiothreitol) = −323 mV; E_0_’(β-mercaptoethanol) = −207 mV; E_0_’(glutathione) = −205 mV; E_0_’(ascorbic acid) = +58 mV [15,16,17]). These findings are further proof that the observed inhibition of rhodesain by curcumin is due to oxidation of thiol groups in the enzyme.

## 3. Materials and Methods

### 3.1. Materials

Recombinantly expressed and purified rhodesain (*T. brucei* cathepsin L-like protease) was provide by Professor Conor R. Caffrey, Center for Discovery and Innovation in Parasitic Diseases, Skaggs School of Pharmacy and Pharmaceutical Sciences, University of California San Diego, San Diego, CA, USA. Benzyloxycarbonyl-phenylalanyl-arginyl-7-amido-4-methyl coumarin (Z-FR-AMC) was purchased from BIOMOL, Exeter, UK. Curcumin was bought from Sigma-Aldrich, Dorset, UK.

### 3.2. Enzyme Assays

The activity of rhodesain was determined with the fluorogenic substrate Z-FR-AMC in 100 mM citrate, pH 5.0, 2 mM dithiothreitol (measuring buffer). Release of free 7-amino-4-methylcoumarin (AMC) was measured at excitation and emission wavelengths of 360 nm and 460 nm in a BIORAD VersaFluor fluorometer, Watford, UK.

#### 3.2.1. Time Course Assay

In order to determine whether the activity of rhodesain was inhibited in a time-dependent manner by curcumin, the enzyme (7 ng/mL; 0.2 nM) was incubated in the presence of 6 μM curcumin dissolved in DMSO in measuring buffer plus 5 μM Z-FR-AMC. For a negative control, rhodesain was incubated with the same amount of DMSO (2%). For a positive control, the enzyme was incubated with 0.1 μM of the established irreversible inhibitor CAA0255 [4,9].

To determine whether the absence of dithiothreitol had an effect on the inhibitory activity of curcumin, rhodesain (35 ng/mL; 1 nM) was incubated with 6 μM curcumin in measuring buffer plus 5 μM Z-FR-AMC but lacking dithiothreitol. The release of free AMC was recorded every minute over a period of 30 min. Then, different antioxidants (dithiothreitol, cysteine, glutathione, β-mercaptoethanol and ascorbic acid) were added in order to determine whether curcumin-inhibited rhodesain could be reactivated. The release of free AMC was recorded every minute for another period of 30 min.

#### 3.2.2. Determination of Inhibitor Type and K_i_

The inhibitor type of curcumin for rhodesain was determined by kinetic analysis. Purified rhodesain was incubated with varying concentrations of Z-FR-AMC (0.125, 0.167, 0.25 and 0.5 μM) and curcumin (0, 2, 4 and 6 μM) in measuring buffer containing 2% DMSO. The final concentration of rhodesain in the assay was 14 ng/mL (0.4 nM). The release of free AMC was recorded as described above every 30 s for 5 min. The velocity of the reaction (relative fluorescence units (RFU)/min) was calculated by linear interpolation of the data. The inhibitor type and the K_i_ value were graphically determined by Lineweaver–Burk plot and Dixon plot, respectively.

#### 3.2.3. Determination of IC_50_

For determination of the half-maximal inhibitory concentration (IC_50_), purified rhodesain was assayed with 5 μM Z-FR-AMC in measuring buffer containing different concentration of curcumin (twofold dilutions from 32 μM to 1 μM) and 2% DMSO. Controls contained 2% DMSO alone. The final enzyme concentrations in the assays were 7 ng/mL (0.2 nM). After 10 min incubation at room temperature, the release of free AMC was recorded. IC_50_ values were determined by linear interpolation according to the method by Huber and Koella [18].

#### 3.2.4. Reversibility Assay

The effect of solvent on the reversibility of the inhibition of rhodesain by curcumin was tested by measuring the recovery of enzymatic activity after dilution of the incubation mixture. Rhodesain (3.5 μg/mL; 100 nM) was pre-incubated with 100 μM curcumin in the presence of 0.1% or 10% DMSO in measuring buffer for 30 min at room temperature. Then, the mixture was diluted 100-fold into measuring buffer containing 5 μM Z-FR-AMC. After 10 min incubation at room temperature, the release of free AMC was recorded. Controls were pre-incubated under the same conditions but in the absence of curcumin.

## 4. Conclusions

Through enzyme kinetic measurements, it was unequivocally shown that curcumin is a reversible, non-competitive, inhibitor of rhodesain. Additional time course and dilution experiments provided conclusive explanations as to why previously it was mistakenly suggested that curcumin is an irreversible inhibitor of rhodesain. Taken together, this study has once more confirmed that curcumin is a promiscuous molecule that can interact non-specifically with any protein under certain incubation conditions leading to misinterpretation of results.

## Figures and Tables

**Figure 1 molecules-25-00143-f001:**
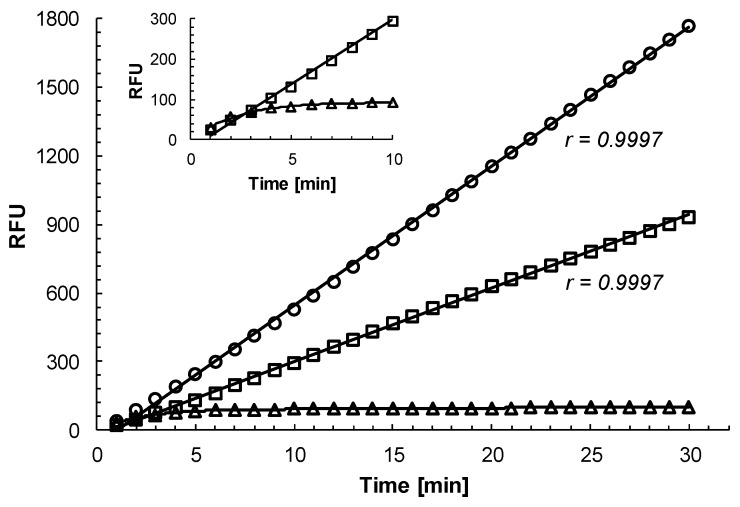
Effect of curcumin and CAA0255 on the substrate hydrolysis activity of rhodesain. Purified rhodesain (7 ng/mL; 0.2 nM) was incubated with 6 μM curcumin (squares), 0.1 μM CAA0255 (triangles), or with DMSO alone (circles) in 100 mM citrate, pH 5.0, 2 mM dithiothreitol, 2% DMSO in the presence of 5 μM of the fluorogenic substrate Z-FR-AMC. The release of free AMC was recorded every minute for 30 min. r, correlation coefficient of the trend line. Insert, enlarged detail view of substrate hydrolysis activity of rhodesain in the presence of curcumin and CAA0255 for the first 10 min. A representative experiment out of three is shown.

**Figure 2 molecules-25-00143-f002:**
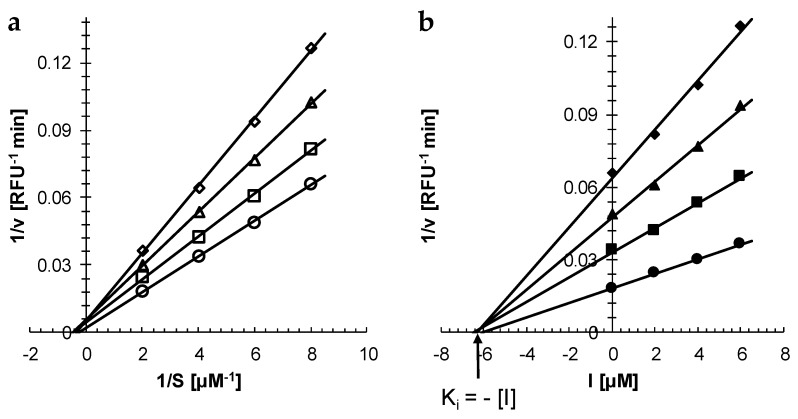
Determination of inhibitor type and constant of curcumin for rhodesain. Purified rhodesain (14 ng/mL = 0.4 nM) was incubated with varying concentrations of Z-FR-AMC (0.125, 0.167, 0.25 and 0.5 μM) and curcumin (0, 2, 4 and 6 μM) in 100 mM citrate, pH 5.0, 2 mM dithiothreitol, 2% DMSO. (**a**) Lineweaver-Burk plot. The concentrations of curcumin were 0 μM (open circles), 2 μM (open squares), 4 μM (open triangles) and 6 μM (open diamonds). (**b**) Dixon plot for determining the inhibitor constant K_i_. The concentrations of the substrate Z-FA-AMC were 0.5 μM (closed circles), 0.25 μM (closed squares), 0.167 μM (closed triangles) and 0.125 μM (closed diamonds). A representative experiment out of three is shown.

**Figure 3 molecules-25-00143-f003:**
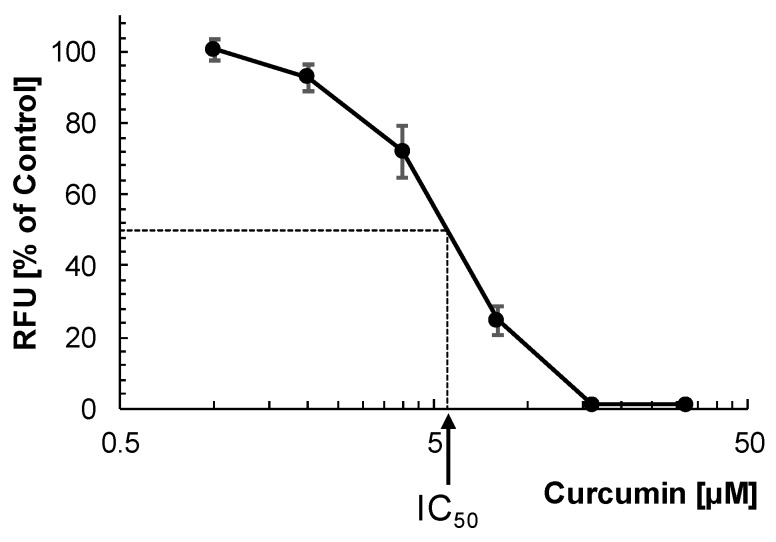
Dose-response curve of the inhibition of rhodesain by curcumin. Purified rhodesain (7 ng/mL; 0.2 nM) was incubated with varying concentrations of curcumin (32, 16, 8, 4, 2 and 1 μM) in 100 mM citrate, pH 5.0, 2 mM dithiothreitol, 2% DMSO, 5 μM Z-FR-AMC. After 10 min, the fluorescence of liberated AMC was measured. The experiment was repeated three times and mean values ± SD are shown.

**Figure 4 molecules-25-00143-f004:**
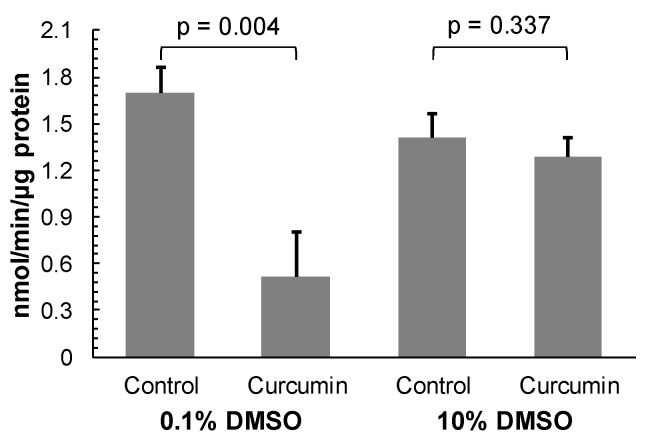
Reversibility of inhibition of rhodesain by curcumin. Purified rhodesain (3.4 μg/mL; 100 nM) was pre-incubated in 100 mM citrate, pH 5.0, 2 mM dithiothreitol with 100 μM curcumin in the presence of 0.1% or 10% DMSO. For controls, the enzyme was incubated under the same conditions but in the absence of curcumin. After 30 min, the mixture was diluted 1:100 into 100 mM citrate, pH 5.0, 2 mM dithiothreitol, 2% DMSO, 5 μM Z-FR-AMC. After 10 min, the release of liberated AMC was recorded. The specific activity (nmol AMC released/min/μg protein) was calculated using a standard curve constructed with uncoupled AMC. Data are mean values ± SD of three experiments.

**Figure 5 molecules-25-00143-f005:**
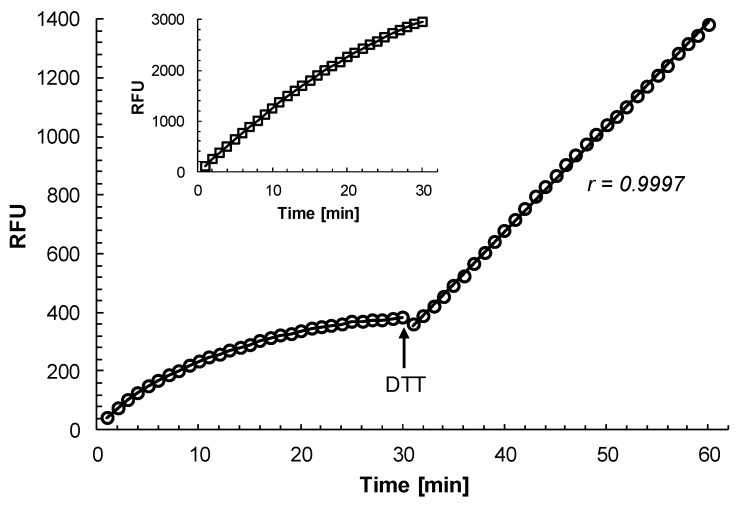
Effect of curcumin on substrate hydrolysis of rhodesain in the absence of the thiol dithiothreitol. Purified rhodesain (35 ng/mL; 1 nM) was incubated with 6 μM curcumin (circles) in 100 mM citrate, pH 5.0, 2% DMSO, in the presence of 5 μM of the fluorogenic substrate Z-FR-AMC. After 30 min, dithiothreitol (DTT) was added to a final concentration of 2 mM (arrow). r, correlation coefficient of the trend line. Insert, substrate hydrolysis of rhodesain in the absence of curcumin and dithiothreitol (squares). The release of free AMC was recorded every minute for 30 and 60 min, respectively. A representative experiment out of three is shown.

**Figure 6 molecules-25-00143-f006:**
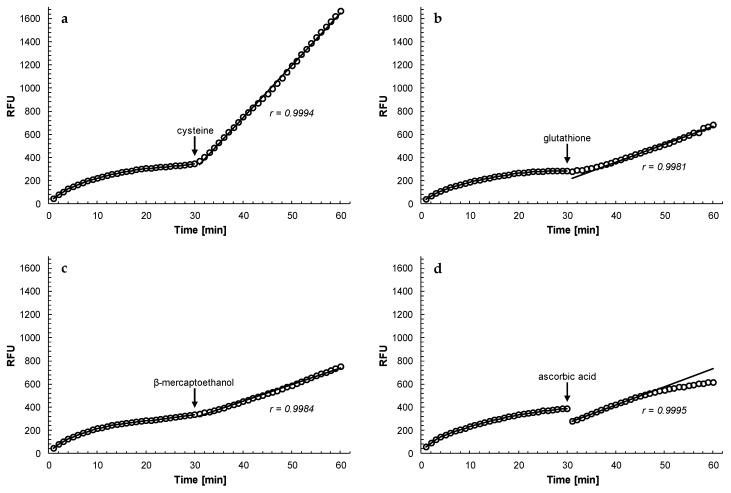
Effect of different antioxidants on the reactivation of rhodesain inhibited by curcumin. Purified rhodesain (35 ng/mL; 1 nM) was incubated with 6 μM curcumin in 100 mM citrate, pH 5.0, 2% DMSO, in the presence of 5 μM of the fluorogenic substrate Z-FR-AMC. After 30 min, antioxidant thiols were added to a final concentration of 4 mM (note that 4 mM of cysteine, glutathione and β-mercaptoethanol (monothiols) equals to 2 mM dithiothreitol (dithiol) based on SH-groups present in the reagents) while ascorbic acid was added to a final concentration of 20 mM (arrows). (**a**) cysteine; (**b**) glutathione (note that the trend line was calculated using the readings from t = 40 min to t = 60 min); (**c**) β-mercaptoethanol; (**d**) ascorbic acid (note that the trend line was calculated using the readings from t = 31 min to t = 45 min). r, correlation coefficient of the trend line. The release of free AMC was recorded every minute for 60 min. A representative experiment out of two or three is shown.

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
