# Peer review of "On the Reversible and Irreversible Inhibition of Rhodesain by Curcumin"

_molecules, 2019, doi:10.3390/molecules25010143_

Round 1
Reviewer 1 Report
A previous study suggested that curcumin acts as an irreversible inhibitor of rhodesain, the major lysosomal cysteine protease of the protozoan parasite Trypanosoma brucei [refs. 3,5]. The author of the current study wrote comments on it previously [ref. 6].
In this manuscript submitted to Molecules, the author provided more data showing that curcumin is a reversible, non-competitive inhibitor of rhodesain, and the apparent irreversible inhibition of curcumin is only observed when no thiol-reducing reagent is included. The data seems strong and convincing, but there are rooms for improvement.
First, the author needs to justify the novelty of the current study, since some reports were in the literature already;
Second, the importance and potential impact to the field should be stated.
Third, the relationship of the current report to Ref. 6 should be clarified. Are there any overlaps?
Author Response
A previous study suggested that curcumin acts as an irreversible inhibitor of rhodesain, the major lysosomal cysteine protease of the protozoan parasite Trypanosoma brucei [refs. 3,5]. The author of the current study wrote comments on it previously [ref. 6].
In this manuscript submitted to Molecules, the author provided more data showing that curcumin is a reversible, non-competitive inhibitor of rhodesain, and the apparent irreversible inhibition of curcumin is only observed when no thiol-reducing reagent is included. The data seems strong and convincing, but there are rooms for improvement.
First, the author needs to justify the novelty of the current study, since some reports were in the literature already;
Second, the importance and potential impact to the field should be stated.
Third, the relationship of the current report to Ref. 6 should be clarified. Are there any overlaps?
Reply: All three comments have been addressed by rewriting the end of the second paragraph of the introduction and the third paragraph has been removed. The revised text reads as follows:
However, it was also recently shown that the inhibition of rhodesain by curcumin seemed to be reversible [6]. A 1:4 dilution of a reaction mixture containing rhodesain and 10 μM curcumin resulted in a 4.7-fold increase in activity [6]. In order to prove conclusively that curcumin is a reversible inhibitor, kinetic studies to determine the inhibitor type of the compound were carried out. In addition, further investigations were conducted to provide explanations for the apparent irreversible inactivation of rhodesain by curcumin observed by Ettari et al. [3,5]. The results of this study revealed that curcumin is a reversible non-competitive inhibitor of rhodesain, a new finding that disproves unequivocally previous claims that curcumin is an irreversible inhibitor. This study also showed that it is important to select the correct assay conditions to measure enzyme activity and to consider the solubility properties of inhibitors otherwise incorrect data will be obtained leading to a misinterpretation of results.
Reviewer 2 Report
This work reflects the author's clear desire, starting from the introduction, to discredit the multiple biological activities of curcumin that have been published by thousands of scientists in recent decades.
Steverding work, on the basis of purely speculative data, also shows an evident intention to demolish the experimental results obtained by Ettari et al (Nat Prod Res 2018, Nat Prod Res 2019):
At page 2, Steverding claims that in the presence of 6 µM of curcumin (a concentration close to IC50) the inhibition of the activity of rhodesain is linear, while when using the irreversible cysteine protease inhibitor CAA0255 the enzyme was quickly completely inhibited.This experiment with these selected doses clearly has no meaning since Steverding used curcumin concentration that inhibited 50% of enzyme activity (i.e. IC50), while for CAA0255 he used a concentration that fully inhibited the enzyme!! So the partial inhibition of rhodesain (i.e. by curcumin) and the full inhibition of rhodesain (i.e. by CAA0255) has nothing to do with the type of inhibition, but only with the selected doses for the two inhibitor.
At page 4 , Steverding claims that the answer to Ettari’s lies is the very low water solubility of curcumin (0.6 µg/mL, i.e. 1.63 µM), thus the concentration selected by Ettari of 50 µM led to a disperded curcumin which specifically inactivate rhodesain.
First of all, it is clear that Steverding does not know that when a study of recovery of enzyme activity has to be carried out, the concentration of the inhibitor that has to be selected is not the IC50, but a concentration around 10-fold IC50 in such a way to fully abolish the enzyme activity, and only after dilution to permit a real evaluation of eventual enzyme-activity recovery (Copeland, Enzymes, Wiley-VCH). In addition it is clear that Steverding speculation in the tentative to demolish Ettari results do not make any sense since in the dose-response curve reported by Ettari (Nat Prod Res 2018) there is an increase of rhodesain inhibition that is dose-dependent: i.e. at 100 µM it was observed 100 % of inhibition vs 90% of inhibition observed at 80 µM, thus highlighting no problem of solubility of curcumin.
Other issue, if for Steverding, according to curcumin solubility (0.6 µg/mL), considered that the maximum soluble concentration for curcumin is 1.63 µM, how did he get the dose-response curve (Figure S3) with the insoluble concentrations of 32, 16, 8 and 4 µM?? There is a clear contradiction between theoretical discussion and adopted procedures in the same paper of Steverding.
At page 4 Steverding claims that the only way to obtain full solubilization of curcumin is to use DMSO concentration up to 10%.
This is clearly incorrect since everybody, working with fluorimetric assays, knows that a high concentration of DMSO can led to a mystification of the fluorimetric assays, since DMSO shows cytotoxic effects (Avicenna J. Med Biochem 2016), so if DMSO is used in high concentration (e.g 10%) there a very high possibility that the inactivation of the enzyme is due not the inhibitor but to the solubilizing agent. So concentration like those used by Steverding up to 10% are unacceptable and clearly mystified the assays.
Lastly at page 5 line 108 again Steverding claims that Ettari experiments did not contain any reducing thiol reagent, which is fundamental to avoid oxidation of the enzyme and gradual inactivation.
Steverding clearly did not even read the experimental procedures of Ettari, since in the supporting information of Ettari (Nat Prod Res 2018) there is clearly written that enzyme buffer contain DTT!
Lastly it is worth nothing that there are an extensive number of literature evidences assessing that curcumin is an irreversible inhibitor, that is not surprising considering that curcumin, due to its two α,β-unsaturated systems is a potent Michael acceptor, thus able to covalently modify several biological proteins containing catalytic SH groups (Nelson et al. 2017; Mathews and Rao, 1991; Awasthi et al., 2000; Fang et al., 2005; Jurrmann et al., 2005).
In view of all above discussed points, since the manuscript contains mystified data is certainly not suitable of publication in this prestigious journal.
Author Response
This work reflects the author's clear desire, starting from the introduction, to discredit the multiple biological activities of curcumin that have been published by thousands of scientists in recent decades. Reply: This is pure speculation, as the reviewer does not know my intention why I carried out this work. I think it is important to repeat and thus to confirm the results of others. In this study, I was not able to reproduce the results of Ettari et al. that curcumin inhibits the activity of rhodesain in a time-dependent manner. Further studies revealed that curcumin is a reversible, non-competitive inhibitor of rhodesain and that the results obtained by Ettari et al. were based on misinterpretations of their data and shortcomings in the experimental set-ups. In addition, there is also a large body of evidence that curcumin is a pan-assay interference compound and an invalid metabolic panacea.
Steverding work, on the basis of purely speculative data, also shows an evident intention to demolish the experimental results obtained by Ettari et al (Nat Prod Res 2018, Nat Prod Res 2019). Reply: First, data cannot be speculative. Data are results obtained from experimental work. Second, I also did not claim that the data published by Ettari et al are wrong; they obtained their data through experimental work. The question is how the data were obtained and interpreted and what the data do tell us. In addition, it is an inappropriate comment to state that I want to demolish the experimental results of Ettari et al. The only thing I want is to set things right. When I was not able to reproduce the results of Ettari et al. and discovered that curcumin is a reversible non-competitive inhibitor of rhodesain, I was still puzzled about the results of Ettari et al. apparently showing that curcumin seemed to be an irreversible inhibitor of rhodesain. Therefore, I continued my studies in order to understand how Ettari et al. got their results.
At page 2, Steverding claims that in the presence of 6 µM of curcumin (a concentration close to IC50) the inhibition of the activity of rhodesain is linear, while when using the irreversible cysteine protease inhibitor CAA0255 the enzyme was quickly completely inhibited. Reply: This is not a claim, it is a fact as clearly demonstrated in Figure 1. This experiment with these selected doses clearly has no meaning since Steverding used curcumin concentration that inhibited 50% of enzyme activity (i.e. IC50), while for CAA0255 he used a concentration that fully inhibited the enzyme!! Reply: For both inhibitors, concentrations close to their respective IC50 values were selected. This is clearly stated in the manuscript (see lines 50-51 and 57-58). It seems that the reviewer did not read the manuscript carefully. So the partial inhibition of rhodesain (i.e. by curcumin) and the full inhibition of rhodesain (i.e. by CAA0255) has nothing to do with the type of inhibition, but only with the selected doses for the two inhibitor. Reply: This is incorrect. The observed inhibition pattern of rhodesain by curcumin and CAA0225 are only because one inhibitor is a reversible inhibitor while the other one is an irreversible inhibitor. It seems that reviewer does not understand the difference between a reversible and an irreversible inhibitor. In addition, if the reviewer would have looked at the insert of Figure 1, they would have seen that for the first two minutes of inhibition, curcumin inhibited the enzyme more potent than CAA0255. After two minutes, the activity of the enzyme is further inhibited by CAA0255 in a time-dependent manner until it is completely inhibited, which is typical for an irreversible inhibitor, while the inhibition of the enzyme by curcumin was time-independent and showed over the whole incubation time the same activity, which is typical for a reversible inhibitor.
At page 4 , Steverding claims that the answer to Ettari’s lies is the very low water solubility of curcumin (0.6 µg/mL, i.e. 1.63 µM), thus the concentration selected by Ettari of 50 µM led to a disperded curcumin which specifically inactivate rhodesain. First of all, it is clear that Steverding does not know that when a study of recovery of enzyme activity has to be carried out, the concentration of the inhibitor that has to be selected is not the IC50, but a concentration around 10-fold IC50 in such a way to fully abolish the enzyme activity, and only after dilution to permit a real evaluation of eventual enzyme-activity recovery (Copeland, Enzymes, Wiley-VCH). Reply: This is again an inappropriate comment to state that I do not know how to carry out a study of recovery of enzyme activity. In the manuscript, I did not suggest that a recovery study of enzyme activity should be carried out at a concentration equal to the IC50 value. As clearly described in the Materials and Methods section and in the legend to Figure 4, I incubated rhodesian with 100 μM curcumin in the presence of different concentration of DMSO. Then, after 30 minutes of incubation, the enzyme solution was diluted into measuring buffer to determine the activity of the enzyme. In addition it is clear that Steverding speculation in the tentative to demolish Ettari results Reply: The reviewer seems to be obsessed of accusing me that I want to demolish the results by Ettari et al. do not make any sense since in the dose-response curve reported by Ettari (Nat Prod Res 2018) there is an increase of rhodesain inhibition that is dose-dependent: i.e. at 100 µM it was observed 100 % of inhibition vs 90% of inhibition observed at 80 µM, thus highlighting no problem of solubility of curcumin. Reply: The reviewer is denying proven evidence that curcumin has a very limited water solubility. The observation that with increasing curcumin concentration rhodesain is inhibited in a dose-dependent manner does not proof that there is no problem with the solubility of curcumin. Other issue, if for Steverding, according to curcumin solubility (0.6 µg/mL), considered that the maximum soluble concentration for curcumin is 1.63 µM, how did he get the dose-response curve (Figure S3) with the insoluble concentrations of 32, 16, 8 and 4 µM?? There is a clear contradiction between theoretical discussion and adopted procedures in the same paper of Steverding. Reply: The reviewer is deliberately ignoring the fact that in my experiment determining the IC50 value of curcumin for rhodesain DMSO at a concentration of 2% was always present in the measuring buffer. This is clearly stated in the Materials and Methods section and in the legend to Figure 3. The presence of 2% DMSO guaranteed to keep curcumin solubilised at the concentration tested.
At page 4 Steverding claims that the only way to obtain full solubilization of curcumin is to use DMSO concentration up to 10%. This is clearly incorrect since everybody, working with fluorimetric assays, knows that a high concentration of DMSO can led to a mystification of the fluorimetric assays Reply: It is unclear what the reviewer means by “mystification of the fluorometic assay”. It may be true that a high DMSO concentration could have a negative effect on an assay using fluorogenic substrates. However, in the case here, I did not use 10% DMSO in the fluorometic enzyme assay. It is clearly stated in the Materials and Methods section and in the figure legends that the concentration of DMSO was always 2% in the fluorometric enzyme assay. In addition, all the controls contained the same amount of DMSO (2%), which had no negative effect on the activity of rhodesain. Again, it seems that the reviewer did not read carefully the manuscript, since DMSO shows cytotoxic effects (Avicenna J. Med Biochem 2016) Reply: It is unclear what the author means with “DMSO shows cytotoxic effects”. It is true that DMSO displays cytotoxic effects but I did not incubate cells with 10% DMSO but purified rhodesain, so if DMSO is used in high concentration (e.g 10%) there a very high possibility that the inactivation of the enzyme is due not the inhibitor but to the solubilizing agent. Reply: On the contrary, I clearly demonstrated in my experiment described on page 4 and in Figure 4 that using 10% DMSO prevented the inactivation of rhodesain when incubated with 100 μM curcumin. So concentration like those used by Steverding up to 10% are unacceptable and clearly mystified the assays. This statement is just nonsense. What does the reviewer mean with “mystified the assays”?
Lastly at page 5 line 108 again Steverding claims that Ettari experiments did not contain any reducing thiol reagent, which is fundamental to avoid oxidation of the enzyme and gradual inactivation. Steverding clearly did not even read the experimental procedures of Ettari, since in the supporting information of Ettari (Nat Prod Res 2018) there is clearly written that enzyme buffer contain DTT! Reply: Again, it is an inappropriate comment to state that I did not read the experimental section of the work by Ettari et al. On the contrary, it seems that the reviewer did not read carefully the experimental section. In the experimental section it is clearly stated “The assay buffer contains: 50 mM sodium acetate, pH = 5.5, 5 mM EDTA, 200 mM NaCl and 0.005% Brij 35 to avoid aggregation and wrong-positive results”. There is no mentioning that DTT was included in the assay buffer. In the next sentence, Ettari et al. indeed stated that “Enzyme buffer contains 5 mM DTT rather than Brij 35” but it is not explained for what the enzyme buffer was used.
Lastly it is worth nothing that there are an extensive number of literature evidences assessing that curcumin is an irreversible inhibitor, that is not surprising considering that curcumin, due to its two α,β-unsaturated systems is a potent Michael acceptor, thus able to covalently modify several biological proteins containing catalytic SH groups (Nelson et al. 2017; Mathews and Rao, 1991; Awasthi et al., 2000; Fang et al., 2005; Jurrmann et al., 2005). Reply: It may be true that there is a number of publications showing that curcumin is an irreversible inhibitor of certain enzymes, but this does not necessarily mean that curcumin must also be an irreversible inhibitor of rhodesain. In fact, none of the examples showed that the inhibition by curcumin was due to a reaction with a catalytic SH-group. Nelson et al. 2017 is a critical review on the biomedical exploration of curcumin. Mathews & Rao 1991 and Awasthi et al. 2000 investigated the reaction of curcumin with glutathione. Fang et al. 2005 studied the reaction of curcumin with selenocysteine in thioredoxin reductase and is the only publication showing that curcumin covalently reacts with an active site amino acid residue, which, however, was not a cysteine residue. Jurrmann et al. 2005 showed the modification of thiols of the interleukin-1 receptor associated kinase. None of the example given included cysteine proteases. Therefore, the argument that there are examples in the literature describing curcumin as an irreversible does not have any relevance for the inhibition mechanism of curcumin on rhodesain.
In view of all above discussed points, since the manuscript contains mystified data is certainly not suitable of publication in this prestigious journal. Reply: The reviewer deliberately misinterpreted the results of my study. The reviewer referred also in a factual incorrect way to my findings. By using terms like “speculative data”, “mystified assay”, mystified data” and “mystification”, the reviewer tried to discredit my manuscript. This also shows that all of the reviewer’s comments are inconclusive and illogical. In addition, the reviewer did not provide any constructive criticism that could be used to improve the manuscript. Therefore, the whole report by this reviewer should be ignored.
Reviewer 3 Report
The work clarifies the inhibitory activity of curcumin with rhodesain and serves to correct a misinterpretation of previous data. Publication without revision is recommended. I found only two typographical errors that can be corrected at the editorial level.
Line 248: non-specifically
Line 275: survival
Author Response
The work clarifies the inhibitory activity of curcumin with rhodesain and serves to correct a misinterpretation of previous data. Publication without revision is recommended. I found only two typographical errors that can be corrected at the editorial level.
Line 248: non-specifically. Reply: typing error has been corrected.
Line 275: survival. Reply: typing error has been corrected.
Reviewer 4 Report
Steverding demonstrates the non-irreversible non-competitive nature of the inhibition of rhodesain by curcumin.
I have a few suggestions:
the sentence "Having shown that curcumin is a reversible" is improperly located. The demonstration of the reversibility of the inhibition is presented after this sentence, in the last part of the manuscript.
The caption of Fig 1A, "A blank recorded with 200
260 μl of 100 mM citrate, pH 5.0 was subtracted from the absorbance values". Why the blank doesn't include DMSO?
Author Response
Steverding demonstrates the non-irreversible non-competitive nature of the inhibition of rhodesain by curcumin. I have a few suggestions:
The sentence "Having shown that curcumin is a reversible" is improperly located. The demonstration of the reversibility of the inhibition is presented after this sentence, in the last part of the manuscript. Reply: I think that this sentence is properly located as the results described in the previous paragraphs and shown in Figures 1-3 clearly demonstrated that curcumin is a reversible, non-competitive inhibitor. Please note that a non-competitive inhibitor is always also a reversible inhibitor.
The caption of Fig 1A, "A blank recorded with 200 μl of 100 mM citrate, pH 5.0 was subtracted from the absorbance values". Why the blank doesn't include DMSO? Reply: DMSO was not included in the blank as the three tests contained different concentrations of DMSO (0.1%, 1% and 10%). The value of the blank of 0.035 that was subtracted is now mentioned in the legend of Figure A1.